# THINK OUT LOUD, PAUSE IN SILENCE: CONFIDENCE-GUIDED REFLECT–PAUSE–ABORT FOR ROBUST AUDIO PERCEPTUAL UNDERSTANDING

## ABSTRACT

Large Audio Language Models (LALMs) mainly fail for two errors: perceptual errors. To address these issues, we propose an adaptive framework that couples perceptual grounding with computation that expands only when needed. First, we introduce PAQA, a Perceptually grounded Audio QA dataset of 7,470 multiple-choice items that pairs multi-speaker, background-rich audio with step-wise reasoning and reflection annotations, enabling supervision of verifiable audio-grounded rationales. On the modeling side, we propose ConfAudio, which unifies explicit, reflective reasoning (fine-tuned on PAQA) with implicit, pause-driven latent computation trained via GRPO. A confidence-aware controller monitors lowest-group-confidence (LGC) during decoding to insert pauses when uncertainty rises and to abort unstable trajectories, thereby reallocating compute toward hard perceptual segments. To stabilize the training process, we design a composite reward that balances answer correctness, reasoning–answer consistency with perceptual robustness, and output format. Across PAQA, MMAU-mini, and MMAR, ConfAudio consistently improves both accuracy and consistency, particularly in noisy, multi-speaker conditions. Our results demonstrate that confidence-guided, adaptive reasoning—grounded in verifiable acoustic evidence—mitigates the dominant perceptual and reasoning failure modes in Audio-QA.

## 1 INTRODUCTION

Large language models (LLMs) have made notable progress in reasoning via chain-of-thought (CoT) prompting and reinforcement-learning (RL) post-training (OpenAI, 2024; Guo et al., 2025), and similar advances have extended to visual modalities (Huang et al., 2025; Feng et al., 2025). Unlike text, audio introduces unique challenges such as overlapping speakers, pronoun ambiguity, shifting emotions, and variable, noisy acoustic conditions. These factors often induce perceptual errors that are among the most prevalent failure modes of current models (Ma et al., 2025b).

Recent audio-capable LLMs (e.g., Qwen2-Audio (Chu et al., 2024), Audio Flamingo (Kong et al., 2024), SALMONN (Tang et al., 2024)) still tend to address audio question answering (Audio-QA) by mapping transcripts directly to answers, with limited verification against the underlying acoustic evidence. Prior audio CoT efforts (Xie et al., 2025; Ma et al., 2025a) supervise long free-form rationales but do not consistently yield improvements on challenging problems. Moreover, RL-only pipelines (Li et al., 2025b; Xie et al., 2025; Wu et al., 2025; Zhong et al., 2025) improve answer accuracy, yet the explicit reasoning process itself has not shown consistent benefits for Audio-QA.

Previous work (Ma et al., 2025b) shows that dominant failures on the MMAR benchmark arise from perceptual errors and downstream reasoning mistakes. This underscores the need to first establish a strong perceptual foundation by explicitly incorporating verifiable evidence, especially in two high-frequency scenarios: (i) distinguishing environmental sounds, and (ii) accurately transcribing multi-speaker conditions. Importantly for audio reasoning, many acoustic cues (e.g., rhythmic density, timbre) cannot be faithfully translated into free-form text, so enforcing text-only rationales risks losing critical granularity. In real-world speech comprehension, humans often reflect on their reasoning process and, when uncertain, pause briefly to deliberate before responding. Inspired by these observations, we pose the central research question: *Can an audio–language model be endowed with adaptive deep reasoning, combining perceptually grounded and verifiable CoT with pause-driven latent computation, to robustly overcome audio perceptual errors?*

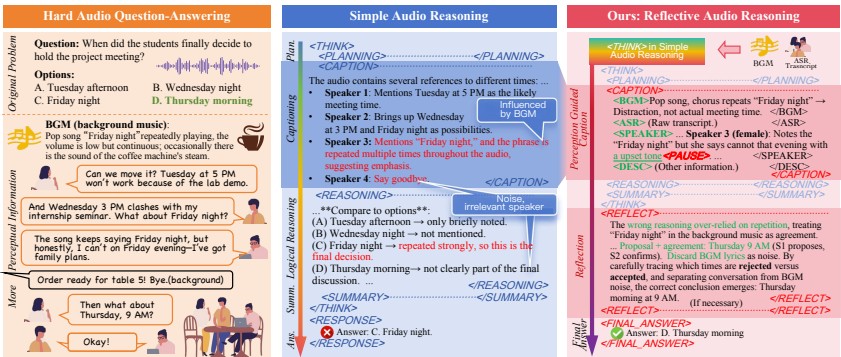

Figure 1: A challenging audio QA case with **perceptual errors**. Repeated lyrics in the background sound/music and multi-speaker overlaps mislead simple reasoning. Although the **CAPTION** includes background sound and cues, the model reaches the correct answer after **REFLECT** reasoning which takes advantage of perceptual information. In particular, when enough acoustic features are encountered, the decoder emits a special **PAUSE** token to allocate additional latent compute.

Interference from background sound and disambiguating multiple speakers are central challenges in real-world audio understanding, yet they remain underrepresented in existing datasets. To address this gap, we construct **PAQA**, the first **P**erceptually grounded **A**udio **QA** dataset, coupling multi-speaker, background-rich audio with stepwise reasoning and reflection annotations. PAQA contains 7,470 multiple-choice Audio-QA pairs. Each example provides structured annotations—including background-music separation, speaker analysis, and multi-turn reflections—and records both an internal answer and a final response. As illustrated in Fig. 1, a model with simple reasoning incorrectly selects "Friday night" due to overreliance on repeated mentions, failing to distinguish background sound from actual conversation. In contrast, a model equipped with reflective reasoning isolates background sound and irrelevant speakers and aligns evidence to arrive at the correct answer.

To handle non-textualizable acoustic signals, we propose **ConfAudio**, an adaptive reasoning framework that unifies explicit reflective reasoning with implicit latent computation. Explicit reasoning is realized through fine-tuning on our curated dataset, while implicit reasoning is enabled by a learned `<pause>` token trained with Group Relative Policy Optimization (GRPO). A confidence-aware module monitors the lowest group confidence (LGC) during decoding and adaptively inserts a pause token when confidence falls below a threshold; it also aborts unstable trajectories to ensure stability. This mechanism encourages the model to attend more deeply to acoustic evidence and to re-segment speakers when necessary. In addition, we design a composite reward that jointly balances answer correctness, reasoning consistency, and output format. We evaluate ConfAudio on the test sets of PAQA, MMAU-mini, and MMAR, where it consistently outperforms strong baselines.

(1) We construct **PAQA**, the first dataset that integrates multi-speaker and background-rich audio with stepwise reasoning and reflection annotations. It covers multi-speaker QA, noisy speech-to-text translation, and environment-centric QA, encouraging models to ground decisions in acoustic evidence rather than transcript shortcuts.

(2) We propose **ConfAudio**, an adaptive reasoning framework that combines explicit audio-grounded reflective reasoning with implicit latent computation. The latter is realized through a learned pause token triggered by a lowest group confidence (LGC) score, enabling the model to silently allocate additional computation when uncertain or abort unstable trajectories.

(3) We design a composite reward that jointly optimizes answer accuracy, format compliance, acoustic consistency, and length control, discouraging unnecessary verbosity. Experiments demonstrate that ConfAudio enhances adaptive reasoning, effectively reducing perceptual and reasoning errors.

## 2 DATA COLLECTION

### 2.1 COMPLEX AUDIO UNDERSTANDING

R1-AQA (Li et al., 2025b) and Omni-R1 (Zhong et al., 2025) show that requiring the model to read *write out* step by step text does not guarantee improvements in Audio QA (AQA). In several AQA settings, explicit CoT provides only marginal or no gains over answer-only decoding, sometimes

even increasing latency. For simple ASR tasks, we also observed that models tend to overfit surface transcripts rather than perform robust reasoning over acoustic evidence (see Fig. 6). Unlike text-only scenarios, audio understanding requires grounding in acoustic cues (e.g. speaker turns, overlapping speech regions) and careful attention to scenarios. Motivated by this, we further analyze Qwen2-Audio's bad cases on the CoTA (Xie et al., 2025) benchmark and identify two major challenges:(1)difficulties in handling multi-speaker conversations, where insufficient speaker diarization under overlapping leads to incorrect attribution of utterances and a consequent loss of dialogue; (2)failures in environmental-sound discrimination, whereby non-speech events and nonsignificant background sound are classified as evidence.

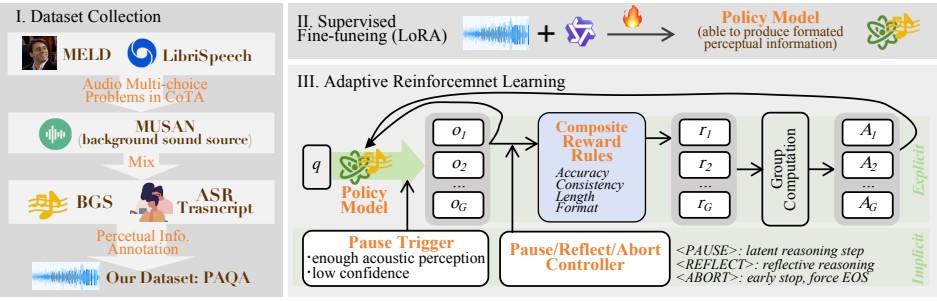

Figure 2: An overview of our work. First, we collected PAQA dataset, with annotating perceptual information like background sound(BGS) and ASR transcript. Secondly, we fine-tuned on PAQA as the policy model in the reinforcement learning. The reinforcement learning mechanism includes pause trigger and controller, and composite reward rules to better performance.

To advance speaker-aware modeling and noise-resilient perception, we construct a dataset that integrates multi-speaker and background-rich audio with stepwise reasoning and reflection annotations.

**Background-rich augmentation** We sample background audio from publicly licensed environmental categories in MUSAN (Snyder et al., 2015) (e.g. alarms, typing, rain, cafeteria, street traffic, soft instrumental music). For a clean speech clip $s$ and a background clip $n$, we first RMS-normalize both and then scale the background so that the power ratio satisfies $\text{SNR}_{dB} = 10$, ensuring that the speech remains ten times stronger than the background, audibly present but not dominant. Each item is annotated with a tag indicating the presence and type of background (e.g., 'Soft instrumental music – please ignore.'), which discourages unnecessary reliance on background cues.

**Multi-speaker Alignment** To discourage models from shortcutting on global transcripts and to encourage speaker-localized reasoning, we annotate turn structures in a `<SPEAKER>` section using a compact, ordered format such as **"Speaker 1: ..."**. We then apply Qwen3-ASR (Team, 2025) to each audio sample to generate a verbatim raw transcript. To mitigate hallucination and drift between summaries and verbatim text, we introduce a quote-presence test (QPT), which measures fuzzy overlap between `<ASR>` snippets $A = a_i$ and `<SPEAKER>` sentences $S = s_i$. Specifically, $\text{SeqRatio}$ is defined as the standard difflib ratio on normalized strings. Items with $QPT < 0.85$ are flagged for revision. The formulation is given by:

$$\text{QPT} = \frac{1}{M} \sum_{i=1}^{M} \max_{1 \leq j \leq N} \text{SeqRatio}\big(\text{norm}(s_i), \text{norm}(a_j)\big). \tag{1}$$

## 2.2 REFLECTION TO CORRECT WRONG INITIAL RESPONSES

In natural conversation, speakers frequently self-monitor and revise their utterances. Building on prior work showing that reflection-driven self-correction improves model performance in reasoning tasks (Shinn et al., 2023; Madaan et al., 2023; Wang et al., 2023), we adopt a reflection-augmented pipeline for complex audio understanding. Concretely, a lightweight baseline model first generates an initial `<RESPONSE>` for each audio-QA item, as illustrated in the third column of Fig. 1. We then automatically detect errors—such as option mismatches, speaker attribution mistakes, hallucinated content inconsistent with ASR transcripts, or misinterpretation of noise cues—and prompt

the model to produce a grounded diagnostic analysis `<REFLECT>`. This analysis explicitly references `<BGM>`, `<SPEAKER>`, and `<ASR>` to explain the failure and localize the supporting evidence. Conditioned on this analysis, the model is guided to generate a corrected `<FINAL_ANSWER>`. For training, we store the triplet (`<RESPONSE>`, `<REFLECT>`, `<FINAL_ANSWER>`), which provides explicit reflection supervision and, from each original audio item, yields an additional corrected example, effectively doubling the supervised data while enriching them with interpretable, perception, grounded self-correction signals. For detailed prompt template, see Appendix C.

The dataset supports a range of tasks, including multi-speaker QA, speech-to-text translation under noise, and environment-centric QA. An in-depth analysis of the final PAQA dataset is provided in Appendix A, while a detailed statistical overview is summarized in Table 1.

Table 1: Dataset Source and Statistics.

| Dataset Source | Main Skills Learning | BGM Used | Quantity | Reflection |
|---|---|---|---|---|
| Multi-Speaker (Xie et al., 2025) | Multi-speaker Speech QA | Free Sound | 1.5k | 1.4k |
| MELD (Poria et al., 2019) | Speech Emotion QA | Sound Bible | 1.5k | 1.4k |
| CoVoST2 (Wang et al., 2020) | Speech-to-Text Translation | No | 1.5k | No |

## 3 METHODOLOGY

### 3.1 FRAMEWORK OVERVIEW

In real-world speech comprehension, humans often reflect on their reasoning process and, when uncertain, pause briefly to deliberate internally before responding (Meyer, 2023). To mirror the sophisticated human ability to process complex audio environments through both explicit explanation and implicit contemplation Boomer (1970); Shriberg (2001), we propose **a dual-mechanism, confidence-gated adaptive reasoning training pipeline**, with an internal confidence score determining the model's reasoning path: an **explicit** Chain-of-Thought with `<REFLECT>`, or an **implicit** latent thinking to handle uncertainty with the `<PAUSE>` token. The whole framework is presented in Fig. 3. Our methodology first employs **Supervised Fine-Tuning (SFT)** to teach the model a specific response structure, compelling it to generate answers that include both an analysis of perceptual audio information and a reflective reasoning process(with `<REFLECT>` in the chain-of-thought). Second, to move beyond simple pattern imitation and enhance the model's adaptive reasoning capabilities, we utilize an advanced reinforcement learning algorithm, **Generalized Reward Policy Optimization (GRPO)**. Within this RL stage, we introduce a key innovation: when low confidence triggers the `<PAUSE>` mechanism, the model activates an **implicit latent reasoning module** to better process ambiguous, non-textual audio cues. This entire process is guided by a specialized reward function designed to optimize the quality and grounding of the final response.

### 3.2 SFT FOR STRUCTURED EXPLICIT GENERATION WITH REFLECTIVE REASONING

Using the specialized dataset detailed in Section 2.2, we explicitly train the model to output a dedicated `<REFLECT>` segment following its initial chain-of-thought reasoning. This structured output ensures that for each prompt, the model produces a transparent analysis of background sound and speaker, and reflection that allows for direct inspection before a final answer is given. We aim to teach the model to generate its responses in a specific, structured format, it lays the groundwork for the subsequent reinforcement learning phase.

### 3.3 RL WITH ADAPTIVE REASONING

#### 3.3.1 GRPO ON FINE-TUNED MODELS

We observed that fine-tuning instruction-tuned models on PAQA leads to performance improvements, although these gains are not significant (see Table 2). While the Supervised Fine-Tuning (SFT) stage effectively establishes a foundation by teaching the model to generate more complex chain-of-thought reasoning, its potential may be inherently constrained by the nature of imitation learning. To transcend these limitations, we introduce a second stage based on GRPO.

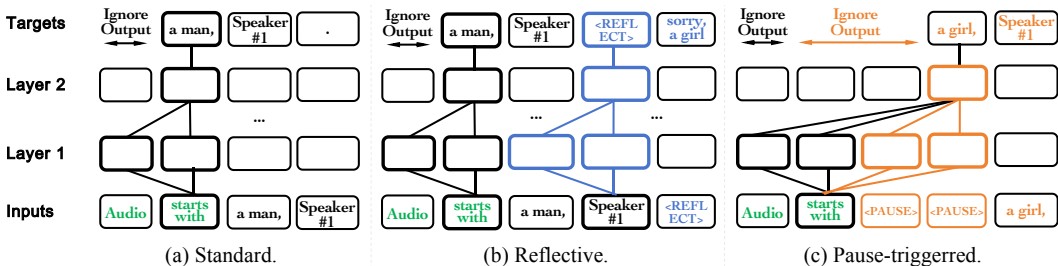

(a) Standard.      (b) Reflective.      (c) Pause-triggerred.

Figure 3: Standard vs. reflective-inference vs. pause-inference (and finetuning). Similar to Goyal et al. (2024), the rounded squares denote one Transformer operation (a self-attention and MLP) in a 2-layer Transformer. Any Ignore Output denotes that during inference, the corresponding output token is not extracted and thus, not fed back autoregressively.

Starting from the SFT checkpoint (reference policy $\pi_{\text{ref}}$ frozen), we generate groupwise rollouts, compute $R(\mathbf{z})$ via equation 8, and update $\pi_\theta$ with GRPO (Shao et al., 2024). We partition rollouts by task group $g \in \{\text{PAQA}, \text{AVQA}\}$ and difficulty bucket (low/med/high). Within each group, we compute groupwise baselines to reduce variance:

$$\tilde{R}^{(i)} = R^{(i)} - \frac{1}{m_g} \sum_{j \in g} R^{(j)}. \tag{2}$$

To process and reason upon non-textual audio cues that are often lost in pure text representations, we incorporate confidence into GRPO by using the lowest group confidence (LGC) as a sample weight (see in 3.3.2). For a trajectory $i$ with task reward $r_i^{\text{task}}$ (covering accuracy, formatting, consistency, and length penalties) and group baseline $\bar{r}$, the advantage is calculated as $A_i = w_i \left(r_i^{\text{task}} - \bar{r}\right)$, where $w_i$ is a clipped, standardized function of the trajectory's LGC, and $w_i = 0$ for filtered samples. The final reward may also include recovery and leak terms when pause control is enabled: $r_i = r_i^{\text{task}} + \eta \cdot \max(0, \text{LGC}^{\text{post}} - \text{LGC}^{\text{pre}}) - \lambda_{\text{leak}} \cdot \mathbf{1}\{\text{leak}\}$, where $\eta, \lambda_{\text{leak}}$ are selected to balance accuracy, formatting, and robustness.

### 3.3.2 IMPLICIT LATENT THINKING

Beyond explicit reflection, the model can also perform hidden computation via implicit pauses. This mechanism is particularly vital when processing complex audio data. Audio streams contain a wealth of perceptual, non-verbal cues—such as speaker intonation, overlapping speech, or ambient noise—that are often difficult to articulate fully in explicit text. An implicit pause allows the model to internally process and weigh these subtle audio features before committing to a textual output, enabling a deeper, more grounded reasoning process.

To govern this behavior, we introduce a confidence-based gating mechanism with two distinct thresholds: $\tau_{\text{pause}}$ for outright termination and a softer threshold $\tau_{\text{pause}} > \tau_{\text{abort}}$. At each decoding step $t$, a specific action is selected based on the model's internal confidence score $C_{k(t)}$:

$$\text{action}(t) = \begin{cases} \text{abort} & \text{if } C_{k(t)} < \tau_{\text{abort}}, \\ \text{pause} & \text{if } \tau_{\text{abort}} \leq C_{k(t)} < \tau_{\text{pause}}, \\ \text{continue} & \text{otherwise.} \end{cases} \tag{3}$$

where `continue` is the default action to normal generation. When confidence falls into the intermediate range, the model triggers an internal deliberation step. When a pause is triggered at $t^\star$, the model emits a `<PAUSE>` special token and generates up to $K_{\text{p}}$ latent tokens $\hat{\mathbf{z}}_{1:L}$ (where $L \leq K_{\text{p}}$). These "pause tokens" are not surfaced in the final visible output and are explicitly excluded from the gradient calculations during parameter optimization. Their sole function is to iteratively update the model's internal hidden state, allowing it to "think" and process complex information before resuming the generation of visible tokens. Formally, the full token sequence and visible output are defined as ($\oplus$ means concatenation):

$$\tilde{\mathbf{y}} = \mathbf{y}_{1:t^\star} \oplus \texttt{<PAUSE>} \oplus \hat{\mathbf{z}}_{1:L}, \qquad \mathbf{y}_{\text{vis}} = \mathbf{y}_{1:t^\star}. \tag{4}$$

To encourage pausing specifically in text involving speech-only, non-textual cues (where extra acoustic reasoning may be beneficial), we maintain a keyword set $\mathcal{A}$, such as "tone", "pitch", whenever any word $w \in \mathcal{A}$ appears in the recent context, we add a positive logit bias $\beta_{\text{ac}} > 0$ to the

<PAUSE> token:

$$\ell_{\text{<PAUSE>}} \leftarrow \ell_{\text{<PAUSE>}} + \beta_{\text{ac}} \cdot \mathbb{I}\big[\exists\, w \in \mathcal{A} \text{ in context}\big]. \tag{5}$$

Token confidence $C_i$ means the negative average log-probability of the top-$k$ tokens at position $i$.

$$C_i = -\frac{1}{k} \sum_{j=1}^{k} log P_i(j) \tag{6}$$

where $k$ denotes the number of top tokens considered. Lower confidence means uncertainty in token prediction. During training, we use lowest group confidence to govern the runtime policy above (pause vs. continue vs. abort), yielding a coherent control scheme. If the confidence score is extremely low ($C_{k(t)} < \tau_{abort}$), it indicates a high likelihood that further deliberation will be unproductive. The abort prevents the model from getting stuck in prolonged, unnecessary pause loops and significantly accelerates overall inference speed by pruning unpromising trajectories early.

### 3.3.3 LOWEST GROUP CONFIDENCE (LGC)

We consider the lowest group confidence metric to provide sufficient signals for estimating the quality of reasoning trajectory. Group confidence averages token confidence over overlapping spans of the reasoning trace. Each token $t$ is linked to a sliding window group $K_i$, consisting of $n$ previous tokens. In particular, we identify its bottom 15% group confidence. For each window, we compute a normalized mean probability:

$$C_{K_i} = \frac{1}{|K_i|} \sum_{t \in K_i} C_t \tag{7}$$

where $|K_i|$ is the number of tokens in group $K_i$. The LGC of the trajectory is then defined as the minimum of these window confidence scores, $\text{LGC}(\mathbf{y}) = \min_{k=1,\ldots,K} C_{K_i}$. This definition emphasizes the weakest local segment within the reasoning trajectory: even a small cluster of highly uncertain tokens can significantly reduce LGC, making it a sensitive indicator of detecting local reasoning collapse, a phenomenon effectively demonstrated by Fu et al. (2025).

## 3.4 COMPOSITE REWARD FUNCTION

To produce correct answers but also generate interpretable and well-organized reasoning, we introduce a novel composite reward function $R$ defined as the weighted sum of four components: besides the accuracy reward and a format reward, we designed a consistency reward for reasoning coherence and explanation quality, and length reward when accurate for stablize the training.

$$R = w_{\text{acc}}\,\mathcal{R}_{\text{acc}} + w_{\text{fmt}}\,\mathcal{R}_{\text{fmt}} + w_{\text{cons}}\,\mathcal{R}_{\text{cons}}(\hat{y}, \hat{y}_{\text{CoT}}) + w_{\text{len}}\,(\mathcal{R}_{\text{acc}} \times \mathcal{R}_{\text{len}}), \tag{8}$$

where $\mathcal{R}_{\text{acc}}$ measures the correctness of the final answer, the format reward, denoted as $\mathcal{R}_{\text{fmt}}$, validates whether the output follows the required structured reasoning schema, $\mathcal{R}_{\text{cons}}(\hat{y}, \hat{y}_{\text{CoT}})$ measures the logical consistency between the final answer and the intermediate reasoning, and $\mathcal{R}_{\text{len}}$ penalizes both under- and over-elaboration, where too short responses are heavily penalized with a score of 0 to reflect insufficient reasoning. Conversely, responses that are too long are penalized using a linear decay function to discourage excessive reasoning.

### 3.4.1 ACCURACY REWARD

The accuracy reward, $\mathcal{R}_{\text{acc}}$, measures the correctness of the final answer. We parse the completion x for the <FINAL_ANSWER> (FA) tag, falling back to the <RESPONSE> (R) tag if FA is absent. Let a completion be denoted by $x$. From $x$, we extract the predicted answer $\hat{y}_{(x)} \in \{A, B, C, D\}$ and compare it to the ground truth label.

### 3.4.2 FORMAT REWARD

The Format Reward validates whether the output follows the required structured reasoning schema; it is assigned 1.0 if all required tags are present, correctly ordered, and properly nested, and 0.0 otherwise. This reward component addresses a common challenge in Reinforcement Learning (RL) known as reward sparsity. We distinguish between two levels of formatting:

Strict Format (Ideal Schema): This is the ideal structure to follow a complex schema. However, enforcing this strict format during RL training is problematic, resulting in a persistent $\mathcal{R}_{\text{fmt}} = 0.0$, providing no gradient for the model to learn the format.

Weak Format (RL Reward for Latent Reasoning): To overcome reward sparsity, $\mathcal{R}_{\text{fmt}}$ is implemented based on a "weak format". This "weak format" approach allows the model to first learn the fundamental `<THINK>`...`<RESPONSE>` sequence. Subsequently, other rewards, such as the consistency reward, implicitly incentivize the model to correctly populate the CoT with the necessary analysis (e.g., ASR, Speaker analysis) to maximize its score, thereby progressively guiding the model toward the ideal "strict format".

### 3.4.3 CONSISTENCY REWARD

Beyond task accuracy and format concerns, we regularize chains for *internal consistency* along three axes: (i) **BGM robustness** blocks spurious causal use of background sound/music; (ii) **Speaker–ASR fidelity** rewards quotes/snippets that appear in the ASR transcript; and (iii) **Reasoning–Answer consistency** rewards agreement between the last internal choice from `<THINK>` and the final answer. Let $r_{\text{bgs}}(x) \in \{0, 1\}$, $r_{\text{spk}}(x) \in [0, 1]$, and $r_{\text{ra}}(\hat{y}, \hat{y}_{\text{CoT}}) = \mathbf{1}\big[\tilde{y}_L(\hat{y}, \hat{y}_{\text{CoT}}) = \hat{y}_L(\hat{y}, \hat{y}_{\text{CoT}})\big] \in \{0, 1\}$. We combine them with a hard background sound gate:

$$\mathcal{R}_{\text{cons}}(\hat{y}, \hat{y}_{\text{CoT}}) = r_{\text{bgs}}(\hat{y}, \hat{y}_{\text{CoT}}) \left( \lambda_{\text{spk}} \, r_{\text{spk}}(\hat{y}, \hat{y}_{\text{CoT}}) + \lambda_{\text{ra}} \, r_{\text{ra}}(\hat{y}, \hat{y}_{\text{CoT}}) \right), \quad \lambda_{\text{spk}}, \lambda_{\text{ra}} \geq 0, \ \lambda_{\text{spk}} + \lambda_{\text{ra}} = 1. \tag{9}$$

In words, any offending background sound/music causal claim zeroes the consistency reward; otherwise, we interpolate between sentence-level Speaker–ASR alignment and self-agreement of the final answer. The default weights $\lambda_{\text{spk}} = \lambda_{\text{ra}} = 0.5$ worked well in our runs.

**Background sound/music robustness** To prevent spurious cues from background sound, we penalize any reasoning/description sentence that *uses background sound/music as causal evidence*. We set $r_{\text{bgs}}(x) = 0$ if reasoning invokes background sound/music as causal evidence (e.g., "bgm", "background sound", instrument names), otherwise $r_{\text{bgs}}(x) = 1$.

**Speaker–ASR fidelity** Within `<THINK>`, let $\mathcal{A} = a_j$ denote ASR sentences and $\mathcal{S} = s_i$ denote speaker snippets (i.e., quoted spans or colon-delimited clauses). We measure how faithfully speaker-attributed quotes in the model's outputs appear in the ASR transcript. Let $\mathcal{S} = \mathcal{S}(\hat{y}, \hat{y}\text{CoT})$ be the set of quoted/snippet spans extracted from the chain (e.g., speaker-attributed quotes), and let $\mathcal{A}$ be the set of sentences from the ASR transcript of the input audio. We softly align each snippet to its best-matching ASR sentence via a normalized similarity $\text{sim}(\cdot, \cdot) \in [0, 1]$, and average over snippets:

$$r\text{spk}(\hat{y}, \hat{y}_{\text{CoT}}) = \frac{1}{|\mathcal{S}|} \sum s \in \mathcal{S} \ \max_{a \in \mathcal{A}} \text{sim}\big(\text{norm}(s), \, \text{norm}(a)\big) \ \in [0, 1]. \tag{10}$$

Here, $\text{norm}(\cdot)$ is a deterministic text normalization (lowercasing, stripping punctuation/speaker tags, collapsing whitespace). The similarity is a normalized edit-based score:

$$\text{sim}(u, v) = 1 - \frac{\text{Lev}(u, v)}{\max\{|u|, |v|\}} \in [0, 1], \tag{11}$$

where $\text{Lev}(\cdot, \cdot)$ is the (character-level) Levenshtein distance.

This rewards quotes that truly occur in the ASR while tolerating minor lexical or punctuation variations; fabricated or hallucinated quotes receive low similarity. For completeness, we adopt the following conventions for edge cases:

$$r_{\text{spk}}(\hat{y}, \hat{y}_{\text{CoT}}) = \begin{cases} 1, & |\mathcal{S}| = 0 \ (\text{no verbatim quotes to verify}); \\ 0, & |\mathcal{A}| = 0 \ \wedge \ |\mathcal{S}| > 0 \ (\text{no ASR to support quotes}); \end{cases} \tag{12}$$

**Reasoning–Answer consistency** In long, noisy chains, decoding drift can make the model "reason to $A$ but output $C$". Therefore, we design rewarding self-agreement. From `<THINK>` we extract the last declared option $\tilde{y}_L(x)$. Let $\hat{y}_L(x)$ denote the final emitted letter (FA/R). Then we calculate $r_{\text{cons}}(x) = \mathbf{1}\big[\tilde{y}_L(x) = \hat{y}_L(x)\big]$.

Table 2: Performance on MMAU Test-mini (Sakshi et al., 2024) and MMAR (Ma et al., 2025b).

| Method | MMAU Test-mini↑ | | | | MMAR↑ | | | |
|---|---|---|---|---|---|---|---|---|
| | Sound | Music | Speech | Average | Sound | Music | Speech | Average |
| Qwen2-Audio | 61.26 | 53.59 | 48.05 | 54.30 | 33.33 | 24.27 | 32.31 | 30.00 |
| +SFT | 62.76 | 44.61 | 55.86 | 54.41 | 41.82 | 34.95 | 45.92 | 40.90 |
| +GRPO-NoThink | 68.17 | 61.38 | 60.66 | 63.40 | 51.52 | 38.83 | 45.92 | 45.40 |
| +GRPO + CoT | 70.27 | 59.88 | 59.46 | 63.20 | **58.18** | 33.98 | 46.60 | 46.30 |
| +GRPO + ExpCoT | 75.07 | 58.98 | 63.66 | 65.90 | 44.85 | 39.81 | 59.86 | 48.20 |
| & - weak format | 72.97 | 61.08 | 63.96 | 66.00 | 42.42 | 43.69 | 61.22 | 49.10 |
| **Ours (ConfAudio)** | **75.67** | **62.27** | **64.26** | **67.40** | **58.18** | **45.63** | **62.59** | **55.50** |
| Audio-CoT | 62.16 | 55.99 | 56.16 | 58.10 | 35.76 | 25.24 | 34.01 | 31.67 |
| Audio-Reasoner | 60.06 | 64.30 | 60.70 | 61.71 | 43.64 | 33.50 | 32.99 | 36.71 |

Together, the consistency measure discourages spurious reliance on background sound/music, enforces alignment between speaker summaries and ASR evidence, and ensures that the internally declared answer matches the final emitted answer.

### 3.4.4 LENGTH SHAPING WHEN ACCURATE

To encourage adequate evidence gathering (speaker attribution, noise filtering) without overlong chains, we introduce a length sub-reward when the accuracy reward is 1. Let $T(x)$ be a simple token proxy (count of non-whitespace sequences in the whole completion). With thresholds $T_{\min} = 300$, $T_{\max} = 600$, and decay scale $K > 0$, we use a piecewise-linear schedule. To enforce clean outputs, we gate by the absence of post-answer content, where any non-whitespace after `</FINAL_ANSWER>` leads to 0.

## 4 EXPERIMENTS

### 4.1 EXPERIMENTAL SETUP

All experiments fine-tune the same pretrained backbones (Qwen2-Audio-7B-Instruct), using the framework introduced by Li et al. (2025b). Training is conducted with a batch size of 1 per GPU, with by 2 gradient accumulation steps, resulting in an effective total batch size of 16. We adopt a learning rate of $1e - 6$, a temperature of 1.0, and configure the GRPO to sample 8 responses per group with a KL coefficient $\beta$ of 0.1. For models incorporating pause latent thinking mechanism, we set tau_pause_quantile=0.50 that allows up to 8 pauses per sequence with 64 thinking tokens each, plus recovery bonus (0.05) and leak penalty (1.0) for think token containment.

We evaluate six configurations: **SFT**, supervised fine-tuning only with no reasoning schema and no `<pause>`; **GRPO-Nothink**, GRPO post-training that emits answers directly without `<REFLECT>` or `<pause>`; **GRPO+CoT**, GRPO enhanced with thinking before the answer (in the weak format of `<THINK><ANSWER>`); **GRPO+ExpCoT**, GRPO enhanced with explicit `<THINK>` (including `<REFLECT>`) but no `<pause>`; **Ours (ConfAudio)**, GRPO enhanced with the explicit schema and `<pause>`; and **External Baselines** including Audio-Reasoner (Xie et al., 2025) and Audio-Thinker (Wu et al., 2025). We use PAQA (train set) for supervised finetuning. For RL training, we utilize 30,000 augmented samples generated upon the AQVA (Yang et al., 2022) dataset, with each response reformulated into a `<think>...</think><answer>...</answer>` reasoning–answer structure. Models are evaluated on several benchmarks, **PAQA Test**(hard), **MMAU** (Sakshi et al., 2024), and **MMAR** (Ma et al., 2025b), the results are listed below.

### 4.2 MAIN RESULTS

Table 2 summarizes the main results across two aggregated benchmarks (MMAU, MMAR). Several key findings emerge from the experimental analysis. A specific case study is shown in Appendix B.

**SFT. vs. Baseline** Injecting audio perceptual information via SFT proves beneficial, particularly in speech, which constitute the primary focus of our study. However, as the data is heavily skewed toward speech, some degree of domain shift is observed in music, leading to suboptimal results.

**RL vs. SFT** GRPO-NoThink performs better than with SFT, and each introduced module in our method improves the performance well. Although GRPO-NoThink outperforms SFT, its improvement stems mainly from reinforcement rather than genuine reasoning. GRPO alone or with naive CoT prompts fails to ensure stable learning. Only when the model is first SFT-trained on our curated audio-reasoning data, which enhances perceptual understanding and format adherence, and then optimized by GRPO under our explicit CoT design, does it achieve coherent and perceptually grounded reasoning across complex audio scenes.

**Pause mechanism works.** Ours method with explicit reflection and implicit latent reasoning surpasses all other baselines. Notably, it improves the Music subset, offsetting the bad performance of just finetuning. More detailed analysis are provided in Appendix A.5.

**Abort mechanism balances speed with formatting penalties.** While abort (early stopping under high uncertainty/high entropy) improves throughput and latency control, we observe that it tends to output think-only simplified content, yielding very low format rewards. We therefore down-weight the format reward (from 0.5 to 0.1) and make a baseline with a weak format. Even under this setting, ConfAudio continues to outperform alternatives, suggesting that the pause mechanism remains the principal driver of performance gains.

### 4.3 ABLATION STUDY

**Background Sound** As shown in Fig.4(a), the introduction of background sound leads to measurable degrades zero-shot performance. However, this drop is substantially mitigated while explicit "ignore prompt" cues are provided. This validates that our reflection step (Section 3) substantially improve accuracy. In Fig. 4(b), we further compare the effect of varying numbers of reflection turns, moving from 0 to 1 round yields a large accuracy enhancement while keeping outputs mostly clean. However, adding more rounds brings diminishing returns and "overthinking". A similar trade-off is identified in the use of the pause mechanism. Excessive pausing negatively affects performance(see Fig. 5), suggesting that it is suitable to set max pause token between 1 and 3. We also evaluate on the test set of PAQA(see Tab.3), on the category of multi-speaker and MELD (Xie et al., 2025), ConfAudio performs the best. Furthermore, under the challenging setting with background sound at SNR=5dB, a condition that considerably degrades most models, our ConfAudio deteriorates the least, retaining state-of-the-art accuracy and consistency. This resilience is attributed to its pause-driven implicit reasoning and rewards aware of background sound/music.

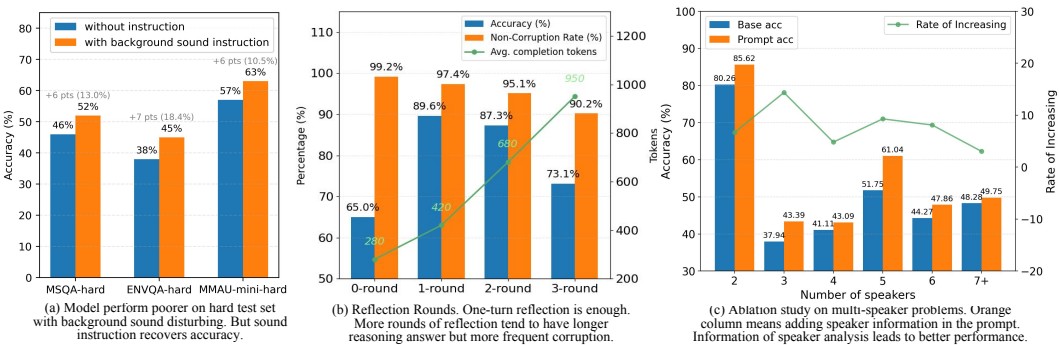

Figure 4: Comparison between different audio situations.

**Multi Speaker** Overall, adding a short "background information" prompt improves accuracy. The benefit is consistent across all speaker counts, and the base model is strong with 2 speakers (80.26%), but drops sharply as speakers increase. This pattern matches the intuition that more speakers introduce attribution and coreference errors. Crucially, 3–6 speaker cases benefit most from the added context. For 7+ speakers, the improvement is modest, indicating that richer cues (explicit diarization tags, role summaries, or brief scene summaries) are likely needed.

## 5 RELATED WORKS

### 5.1 LARGE AUDIO–LANGUAGE MODELS (LALMS)

Early LALMs such as Qwen2-Audio(Chu et al., 2024), Audio Flamingo(Kong et al., 2024), and SALMONN(Tang et al., 2024) advanced ASR, but remained fragile in real-world reasoning tasks in-

| Model | Multi-Speaker(hard) | | BGM-rich Acc. | |
| --- | --- | --- | --- | --- |
| | Acc. | Consistency ↑ | SNR=10 | SNR=5 |
| Qwen2-Audio | 42.2 | 38.5 | 41.0 | 20.1 |
| +SFT | 46.2 | 41.5 | 44.0 | 31.2 |
| +GRPO-NoThink | 52.7 | 48.3 | 50.2 | 38.4 |
| +GRPO-ExpCoT | 61.5 | 58.7 | 60.8 | 47.6 |
| **Ours** | **70.4** | **68.1** | **69.5** | **57.8** |
| Audio-CoT | 50.6 | 46.9 | 48.3 | 35.0 |
| Audio-Reasoner | 56.8 | 52.7 | 55.9 | 41.8 |

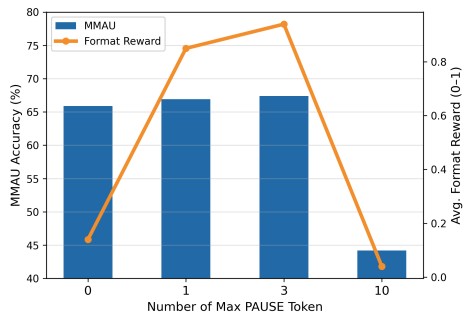

Table 3: Evaluation on the test set of PAQA. Our model performs best in each category.

Figure 5: Abaltion study of #<PAUSE> tokens. Set max pause token as 1-3 is suitable.

volving overlapping speakers and non-stationary noise. On-demand CoT in Audio Flamingo 3(Goel et al., 2025) and structured CoT in Audio-Reasoner (Xie et al., 2025)—yet models often reverted to transcript shortcuts whenever acoustic evidence was difficult to verbalize. To address these limitations, we release a structured dataset that couples multi-speaker and background-rich audio, explicitly guiding LALMs to ground decisions in acoustic rather than purely textual evidence.

### 5.2 Explicit Reasoning in Large Language Models

In LLMs, structured reasoning through CoT, reflection, and RL post-training has yielded consistent gains beyond supervised fine-tuning (SFT) (Guo et al., 2025; Team et al., 2025). While Vision-R1 (Huang et al., 2025) and Video-R1 (Feng et al., 2025) extended RL-based reasoning to over-thinking suppression. In audio, GRPO-style RL underlies R1-AQA and Omni-R1 (Shao et al., 2024; Li et al., 2025b; Zhong et al., 2025), with mixed evidence on whether RL alone suffices. More recent approaches (Wen et al., 2025; Wu et al., 2025; Li et al., 2025a; Jin et al., 2025) highlight that objectives should reward useful and concise reasoning rather than verbosity. In this work, we instead unify explicit, audio-grounded reasoning with reflection, operationalized through a multi-term reward that enforces correctness and conciseness.

### 5.3 Implicit Reasoning and Pause-Gated Latent Compute

Complementary to explicit rationales, implicit computation allocates additional internal processing before token emission. Learned <pause> tokens can trigger silent forward passes (**?**), echoing earlier adaptive-computation approaches(Graves, 2017; Banino et al., 2021) that learn instance-dependent halting policies. To our knowledge, such latent computation has not been systematically validated in audio–language reasoning. Our contribution is to extend <pause> to LALMs and couple it with a lowest-group-confidence (LGC) controller: when confidence drops on acoustically inexpressible cues, ConfAudio diverts into a short, budgeted latent stream and can abort tail trajectories under severe uncertainty.

## 6 Conclusion

In this paper, to address two entangled failure modes in Audio-QA, perceptual and reasoning errors, we build **PAQA** to supervise verifiable, audio-grounded questions-answerings, and propose **ConfAudio**, which couples explicit reflection with implicit, pause-driven latent thinking trained via GRPO with a composite reward. Specially, a lowest-group-confidence controller inserts <pause> or aborts unstable trajectories. ConfAudio delivers consistent gains in accuracy and consistency under noisy, multi-speaker conditions, narrowing the gap between acoustic evidence and reasoning.

### Ethical considerations

Our dataset is constructed from publicly available corpora or controlled augmentations, with all speech either anonymized or synthesized to avoid privacy leakage. Despite the contributions, several limitations remain. First, while our dataset is carefully annotated with multi-speaker and

background-rich reasoning structures, its scale is modest compared to general-purpose audio corpora, which may limit coverage of rare conversational phenomena.

## REPRODUCTIVITY STATEMENT

We prioritize reproducibility by releasing dataset specifications, and preprocessing scripts for background injection, speaker segmentation, and ASR alignment. All training configurations—including optimizer settings, batch sizes, learning rate schedules, and LoRA ranks—are documented and released as YAML files. Our evaluation follows a consistent protocol across our dataset, MMAU, and MMAR, reporting accuracy, consistency, and robustness under noise. Results are averaged over multiple random seeds to avoid cherry-picking. Upon publication, we will release our training data, code, inference pipelines, and checkpoints under an open-source license.

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

## A  DATA COLLECTION OF HARD AUDIO UNDERSTANDING

### A.1  LIMITATIONS OF SIMPLE ASR-CENTRIC TEXT REASONING

Early approaches to audio reasoning typically relied on converting speech into text via automatic speech recognition (ASR) and then performing reasoning over the textual transcript. While effective to some extent, this paradigm inevitably discards information that is uniquely embedded in the audio signal itself. To probe the limitations of this pipeline, we first evaluated the ASR+text reasoning approach on benchmarks such as CoVoST2 and MMAU. In CoVoST2, model performance is largely determined by raw ASR accuracy, and we observed that "simple ASR" signals are quickly memorized without yielding robust generalization. A case study is shown in Fig.8, which highlights several intrinsic challenges. Homophones and proper-name ambiguities necessitate long-range semantic modeling and external knowledge retrieval, while gendered pronouns in Chinese (e.g., "he/she") lack reliable acoustic cues and thus require contextual inference for disambiguation. In particular, Paraformer's frame-level alignment, coupled with strong language model priors, tends to induce a "nearest-neighbor copying" effect—yielding high accuracy on in-distribution transcripts but exhibiting pronounced failures under distributional shifts. Moreover, exposure to translation-oriented data (e.g., CoVoST2) can bias models such as Qwen-Audio to mistakenly trigger translation behavior, sometimes converting Chinese speech into other languages when acoustic cues are uncertain.

In Fig. 7(a), there is an improvement on base models if we asked them to answer questions with thinking in the format of `<THINK>...</THINK><FINAL_ANSWER>...</FINAL_ANSWER>`. Therefore, we collected 2,050 samples from a subset of CoVoST2 (including 50 challenging cases reserved for the test set) and employed Kimi to generate CoT annotations. Using this data, we fine-tuned Qwen2-Audio and evaluated them on the designated test set. However, the models exhibited severe overfitting (see Fig. 6(b)) after only a single epoch of training: while the outputs consistently followed the required `<THINK>...</THINK><FINAL_ANSWER>...</FINAL_ANSWER>` format and the training loss rapidly approached zero, the test accuracy dropped below 5%. This observation indicates that the gradients primarily optimized for surface-level grapheme mapping and fixed output formatting, without fostering genuine cross-sentence reasoning, coreference resolution, or knowledge-grounded inference.

Consequently, these observations indicate that the "Thinking" component of chain-of-thought supervision should be allocated primarily to more challenging audio understanding tasks, such as multi-speaker dialogues and noisy environments—where reasoning signals genuinely drive the model to overcome semantic ambiguities and enforce knowledge-aware interpretations, rather than merely replicating templates on simple ASR tasks.

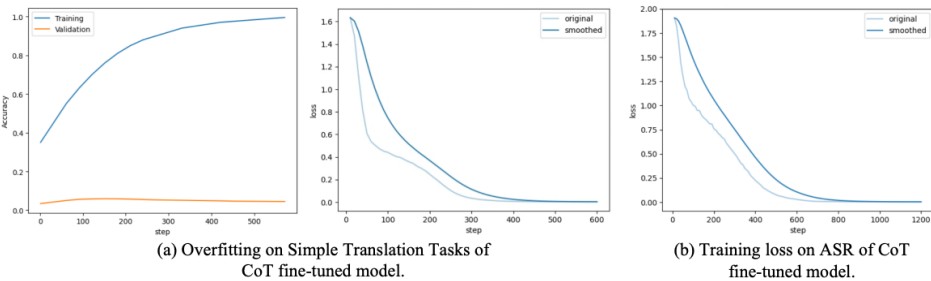

(a) Overfitting on Simple Translation Tasks of CoT fine-tuned model.

(b) Training loss on ASR of CoT fine-tuned model.

Figure 6: The training dynamics of a chain-of-thought (CoT) fine-tuned model (Qwen2-Audio-7B), indicating the model overfits to the training set in simple translation tasks. This suggests that CoT fine-tuning without additional regularization or more diverse data fails to yield robust generalization, particularly for tasks requiring broader reasoning beyond surface transcript matching.

### A.2  HARDER AUDIO REASONING TASKS

### A.3  SYNTHETIC AUDIO WITH BACKGROUND SOUND

Following this, we further analyzed erroneous predictions of Qwen2-Audio on the MMAU benchmark. As shown in Fig.7(b), we compared fine-tuning trajectories on the MSQA dataset with and

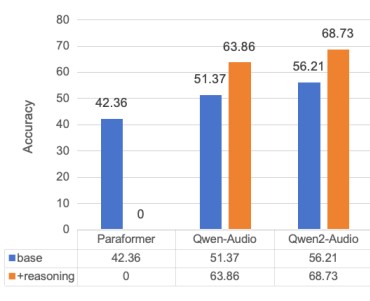 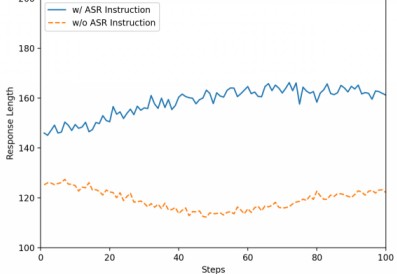

(a) the comparison of base model and model with reasoning prompt.

(b) the comparison of prompt with or without ASR finetuned on MSQA.

Figure 7: Experiments on the Exploration of Good Audio Reasoning prompt.

| English Answer | Chinese Answer | Paraformer | Qwen-Audio | Qwen2-Audio | Qwen2-Audio+ BadThought | Analysis |
|---|---|---|---|---|---|---|
| Cuella | 库艾拉 | 屈埃拉 | Quoi elle. | Quoi elle. | 屈埃拉 | For very short examples, Qwen2-Audio and Qwen-Audio will do translation automatically. |
| Mill Creek is an unincorporated area located in Pope County, Arkansas, United States. | 米尔克里克市是位于美国阿肯色州波普县的一个非建殖地区。 | 米尔克里克市位于美国阿肯色州波普县的一个非建殖力区 | 米尔克里克市是位于美国阿肯色州波普县的一个非建制郡。 | 米尔克里克市是位于美国阿肯色州波普县的一个非建制地区。 | 米尔克里克市是位于美国阿肯色州波普县的一个非建置力区。 | (1) Homophones are difficult to distinguish; (2) The results of the qwen2audio model are easily misled by bad answers; The term "unincorporated county" is semantically contradictory; "Bopu County" has clearly defined its administrative level as "county", so its subordinate cannot be a "county". |
| Caul fat is a thin membrane surrounding the internal organs of animals, used as a cooking ingredient. | 网油是围绕动物内部器官的薄膜为烹饪食材 | 网游是围绕动物吴内部器官的薄膜为烹饪食材 | 网油是围绕动物内部器官的薄膜为烹饪食材 | 网油是围绕动物内部器官的薄膜为烹饪食材 | 网油是围绕物内部器官的薄膜为烹饪食材（模型输出正确，无需修正） | Qwen2-Audio and Qwen-Audio are better at capturing contextual information compared to Paraformer. |
| Night scenery is really hard to capture | 夜景真的超难拍 | 液晶真的超难拍 | 液晶真的超难拍 | 液晶真的超难拍 | 液晶真的超难拍 | But hard cases still cannot be accurately identified. "Night scene" and "liquid crystal" have the same pronunciation, but the former is more suitable for the context of "taking photos". |

Figure 8: Bad cases of Paraformer, Qwen-Audio and Qwen2-Audio on the simple ASR task.

without ASR-augmented data. The results reveal that models trained with ASR supervision exhibit substantially longer response lengths, which we interpret as a proxy for deeper and more structured reasoning ability. This finding suggests that integrating ASR data into training not only improves transcription accuracy but also enhances the reasoning capacity of audio-language models. Therefore, in the first stage of fine-tuning, we deliberately incorporated the ASR-enriched data described in the previous section to further consolidate the model's ASR capability as a foundation for downstream reasoning.

Moreover, we processed the audio with MUSAN(Snyder et al., 2015), which satisfies target 10 dB SNR, according to

$$\text{SNRdB} = 10 \log 10 \left( \frac{P_s}{P_{n,\text{scaled}}} \right) = 10.$$

Let $P_s = \frac{1}{T} \sum_t s_t^2$ and $P_n = \frac{1}{T} \sum_t n_t^2$. The background gain is

$$k = \sqrt{\frac{P_s}{P_n \cdot 10^{\text{SNR}_{\text{dB}}/10}}} = \sqrt{\frac{P_s}{P_n \cdot 10}}.$$

## A.4 Audio Question-Answering with Multi Speakers

We use the subset of Multi-Speaker Dataset in CoTA (Xie et al., 2025), which is generated by TTS to navigate intricate speaker interactions. First, we generated diverse conversational texts with LLMs. Next, using timbres from LibriSpeech as prompts, we synthesized high-quality speech via the CosyVoice2 framework. Finally, these distinct speech samples were combined into a rich dataset.

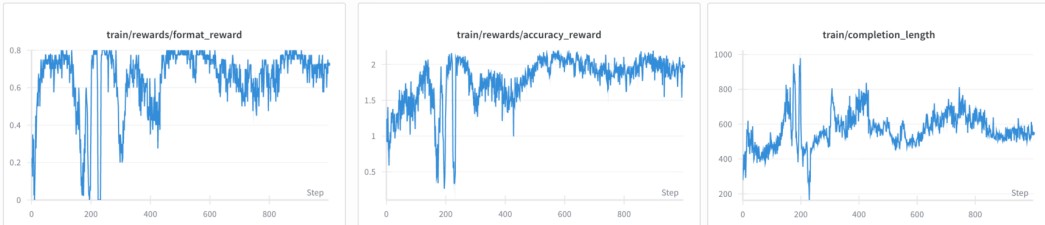

Figure 9: GRPO Training. Overall, the RL training progressed well, but there is a clear collapse around 200 steps. The trigger was the length-reward design: during exploration, longer completions earned higher scores, but once a response exceeded 600 tokens, a linear decay penalty kicked in. The policy reacted by abruptly shortening completions to 200 tokens; these outputs were often incomplete, so the format reward dropped to 0, and the accuracy reward fell to 0.5. After this shock, training recovered and stabilized, indicating the policy adapted to the length constraint.

### A.5 PROPER RESPONSE LENGTH AFTER LATENT REASONING

Though more stable during training, introducing pause-based latent tokens increases training time, raising max_pause_token from 1 to 3 roughly doubles training time. See more details in Fig.9. Therefore, we set a length reward in the design of whole reward function. We also observe some findings about the design of length-reward Sec. 3.4.4. Overall, the RL training progressed well, but there is often a clear performance drop about 200 steps. The instability can be attributed to the length-reward: during RL exploration, the model received higher scores for generating longer responses, but once a response exceeded 600 tokens, a linear decay penalty kicked in. In reaction, the policy abruptly shifted to producing shorter outputs; these truncated responses were often incomplete, leading to a format reward drop to zero and a reduction in accuracy reward to 0.5. Following this disruption, the training process gradually recovered and ultimately stabilized, indicating the policy capacity to adjust its generation in response to complex reward signals (See Fig. 9).

## B CASE STUDY

As shown in Fig.1, the case highlights two failure modes: perceptual misbinding and salience-driven rationale drift. The naive system exhibits this by prioritizing the surface frequency of "Friday" while overlooking its negated polarity and the logical flow of the proposal-to-confirmation sequence. Conversely, the reflective controller rectifies this by enforcing evidence typing (differentiating background sounds from linguistic turns) and ensuring dialogue-act alignment, ultimately restoring causal fidelity to the acoustic evidence.

### B.1 MULTI SPEAKERS TRACKING

> **Bad Case A**
>
> [ASR excerpt] [S1] "Ship date is the 12th if QA passes." [S3] "QA won't finish by the 12th." [S4] "Set the launch to the 15th." [S2] "Not the 5th—I said the 15th." [S1] "Agreed."
> [Question] What is the final launch date?
> [Choices] A 5th · B 12th · C 13th · D 15th
> [Audio context] Meeting room; steady HVAC hum as BGM; frequent interruptions.
> [Ground truth] D
> [Simple model response] B
> [Reflection] <REFLECT> S3 explicitly negates the initial "12th" suggestion, S4 introduces a new proposal "15th," S2 clarifies that it is "15th" rather than "5th," and S1 provides team confirmation; taken together these turns establish the final launch date as the 15th.

This case tests how well the model can follow a conversation in a realistic, noisy meeting. The audio includes a constant low hum from an air conditioner (HVAC) in the background, and the speakers often interrupt each other. These challenges make it difficult to follow the flow of the conversation. The simple model's failure wasn't about hearing the words; it was about failing to understand the process of the discussion. The reflective model succeeded because it built a step-by-

step understanding of the conversation's logic. It correctly handled the disagreements and changes, proving it can reason reliably even in a noisy and dynamic group setting.

## B.2 EXPLICIT REFLECTION

---

**Bad Case B**

```
<THINK>
<PLANNING>
```
The user's question is: Based on the conversation, what are the speakers trying to assemble? I need to analyze the dialogue between the speakers to identify the item they are constructing. The audio provides the audio clip, and I should analyze it for details related to the assembly activity. `</PLANNING>`
`<CAPTION>`
The audio features a conversation among three individuals in what sounds like a workspace. They are discussing an instruction manual and the assembly process. - Speaker 1: Expresses frustration with the instruction manual, stating that it's difficult to distinguish pieces A and B.
- Speaker 2: Proposes that piece B has a slightly beveled edge, but concedes that it's similar.
- Speaker 3: Agrees that it looks like Bevel Edge and expresses hope that it's correct.
- Speaker 1: States it is Bevel Edge B, and declares they can complete the assembly.
- Speaker 2: Hope is right and doesn't want to misassemble again.
- Speaker 1: Confirms it is Bevel Edge B and asks everyone to take deep breaths and move on.

The core task is assembling an item where they are currently putting together parts, such as bevel edges.
`</CAPTION>`
`<REASONING>`
1. Analyzing the Dialogue: The dialogue revolves around distinguishing parts (A and B) and confirming a beveled edge.
2. Identifying the Assembly Activity: The key element is Bevel Edge, suggesting the speakers are assembling something with beveled sides.
3. Evaluating Options: - (a) A table: While a table might require beveling, the central focus isn't just about assembling a table. - (b) A chair: Similarly, a chair might require beveling, but it isn't their primary focus. - (c) A bed: A bed typically features beveled edges, aligning perfectly with the conversation. This is the most likely scenario. - (d) A bookshelf: A bookshelf is less likely to have beveled edges.
4. Concluding the Item: Based on the bevel edges and the distinct focus on distinguishing them, it can be inferred that the speakers are assembling a bed. `</REASONING>`
`<SUMMARY>`
The speakers are differentiated by focusing on the Bevel Edge assembly of a bed. By eliminating other less relevant options through reasoning, the final conclusion is that they are assembling a bed. `</SUMMARY>`
`</THINK>`

---

Here is the reflection of this case.

---

**Reflection of Bad Case A**

`<REFLECT>` Evidence available from the dialogue is generic to many flat-pack builds: "piece A vs. B," "bevel(ed) edge," "don't misassemble," and "let's move on." No category-specific anchors are heard (e.g., for a bed: headboard/footboard/rails/slats; for a bookshelf: shelf/upright/back panel/bracket; for a table: tabletop/apron; for a chair: backrest/seat/armrest). Given absent anchors, the safest policy is to avoid over-commitment. If forced to choose among the four, "bookshelf" is the most compatible with beveled front edges on shelves and frequent A/B part distinctions during shelf panel assembly. Confidence remains low; with an "uncertain" option, I would state insufficient evidence. `</REFLECT>`
`<ANSWER>`Bookshelf.`</ANSWER>`

---

## B.3 PAUSE MECHANISM

This case tests the model's ability to understand meaning that isn't in the words themselves, but in the way they are spoken (the "tone," or prosody). The audio is a phone call with background noise from rain and static (line hiss). The challenge is to see if the model can spot sarcasm or reluctance, even when the words seem positive. Therefore, this case highlights a major flaw in models that only read text. They can be easily fooled when a speaker's tone changes the meaning of their words (like in sarcasm or politeness). The reflective model's success shows it can listen, not just read. It

integrates how something is said with what is said to find the true meaning, which is crucial for understanding human conversation.

---

**Bad Case C**

[ASR excerpt] [S1] "Can you present Friday?" [S2] "Yeah, Friday... great." (F0 downward; reduced energy; elongated final syllable) [S2] "Actually, I'm not available Friday evening." [S1] "Then Thursday morning?"
[Question] Is S2 willing to present on Friday?
[Choices] A Yes · B No · C Unsure · D Only afternoon
[Audio context] steady rain and mild line hiss
[Ground truth] B
[Simple model response] A
[Reflection] <REFLECT> ..."Yeah, Friday... great" ¡PAUSE¿ check tone/intonation. ¡/PAUSE¿ The phrase "Yeah, Friday... great" is produced with a downward F0 contour and reduced energy consistent with reluctance or irony, and a subsequent turn states explicit unavailability on Friday evening; integrating prosodic and semantic cues indicates S2 is not willing to present on Friday.
</REFLECT>

---

In sum, the Multi-speaker Tracking matters speaker attribution ("who"), the Pause Mechanism addresses the transition from rapid, text-centric processing to a more computationally intensive, multi-modal analysis, and Reflection focuses on the ground truth by deploying targeted evidence re-querying, contextual anchoring to disambiguate local hypotheses (as in ASR N-best lists), and integrating conflicting cross-modal data. The convergence of these mechanisms allows the model to emulate human-like cognitive robustness in complex, ambiguous, and multi-participant acoustic environments, distinguishing its performance from that of a passive, deterministic system.

## C PROMPT TEMPLATE

---

**A Sample Case of Our PAQA Data**

```
<THINK>
<PLANNING>
The user wants to understand the dynamic changes ...... </PLANNING>
<CAPTION>
<BGM>Street noise; intermittent car horns.</BGM>
<SPEAKER>A: adult F; B: adult M; alternating turns.</SPEAKER>
<ASR></ASR>
<DESCRIPTION>The audio clip predominantly features static noise. ...... similar to that of a detuned
television or a device failing to receive a signal.</DESCRIPTION>
</CAPTION>
<REASONING>
1. Identify changes in Intensity (Volume): ...... struggling to maintain a consistent output, adding to the
impression of something malfunctioning or broken. </REASONING>
<SUMMARY>
The static noise in the audio is highly dynamic. ... leading to a sense of disorder and instability.
</SUMMARY>
</THINK>
<RESPONSE>
The audio presents a static noise, ...... is one of energetic chaos, preventing any possibility of calm or
predictability. </RESPONSE>

<REFLECT1> Does "A" mention the cake, not B? Check turn 3.</REFLECT1>
<NEW_RESPONSE>A</NEW_RESPONSE>
<REFLECT2> Does "A" mention the cake, not B? Check turn 3.</REFLECT2>
<NEW_RESPONSE>B</NEW_RESPONSE>
```

---

**Prompt template of Refeclection Sample**

After producing the <RESPONSE>, you must perform a structured self-reflection step.
1. Compare the <RESPONSE> with the overall task requirements and check for issues such as: - Missing or incomplete coverage of the audio content (did it stop too early? were some speakers/segments missed?). - Repetition or redundant phrasing that should be removed or marked clearly. - Speaker attribution or diarization errors (wrong speaker assignment, merged speakers, or split speakers). - Prosody/tone/intonation mistakes or overemphasis on irrelevant details. - Inconsistent reasoning or labels (final choice must align with the reasoning and context). - Overly simplistic or single-hypothesis reasoning when alternatives exist.
2. Inside <REFLECT>...</REFLECT>, explicitly list: - The problems found in <RESPONSE>. - The corrections or adjustments needed (without referencing or leaking the gold standard answer text). - Any uncertainties or low-confidence areas.
3. Then rewrite the improved answer inside <FINAL_ANSWER>...</FINAL_ANSWER>, ensuring: - All necessary content is covered. - No hallucinated details are added beyond the given <CAPTION>, <ASR>, and <DESCRIPTION>. - Speaker attributions and reasoning are consistent. - The final answer matches the reasoning and is labeled correctly with confidence if required.
Format strictly as: <REFLECT> [Your structured reflection here] </REFLECT>
<FINAL_ANSWER> [Your corrected, high-quality final answer here] </FINAL_ANSWER>
Here is the original bad answer: Turn0 Here is the golden answer: Golden_Ans

---

## D THE USE OF LARGE LANGUAGE MODELS (LLMS)

In order to reduce typos during the writing process and to optimize complex sentence structures so that the article becomes simpler and easier to read, we use mainstream large language models to refine certain paragraphs. For example, we use prompts such as "Help me correct the typos and grammatical errors in the above text, and streamline the logic to make it clear and easy to understand."

# E  CODE

---

**Algorithm 1 Pseudocode of Pause/Abort/Reflect Mechaiam in ConfAudio.**

---

**Require:** query $q$, audio $a$; policy model $\pi_\theta$, reference model $\pi_{\text{ref}}$; window $K$; max pauses $M$; max length $L$; thresholds (pause $\tau_p$, abort $\tau_a$, and reflect $\tau_r$)

1: **function** $\text{CONF}(\ell, \ell_{\text{ref}}, a)$            ▷ inputs are logits, entropy↓ + KL↓ + audio-uncertainty↓

2:      $\pi \leftarrow \text{softmax}(\ell/T); \quad \pi_{\text{ref}} \leftarrow \text{softmax}(\ell_{\text{ref}}/T)$

3:      $H_{\text{n}} \leftarrow \dfrac{-\sum_{v \in V} \pi_v \log \pi_v}{\log |V|}, KL_{\text{n}} \leftarrow \dfrac{\sum_{v \in V} \pi_v \log \frac{\pi_v}{\pi_{\text{ref},v}}}{\log |V|}$      ▷ normalized entropy, and KL in $[0, 1]$

4:      $u_{\text{n}} \leftarrow \text{norm}\big(u(a)\big) \in [0, 1]$            ▷ audio uncertainty (e.g., SNR/ASR confidence)

5:      **return** $\text{clip}\big(w_1(1-H_{\text{n}}) + w_2(1-KL_{\text{n}}) + w_3(1-u_{\text{n}}),\ 0,\ 1\big)$

6: **end function**

7: **function** $\text{MELTDOWN}(y)$            ▷ e.g., high $n$-gram repeat or abrupt entropy drop

8:      **return** $\big(\text{Repeat}(y) > \gamma\big)$

9: **end function**

10: **function** $\text{CONFAUDIOGENERATE}(q, a)$

11:      $y \leftarrow [\,], h \leftarrow [\,], p \leftarrow 0, low \leftarrow 0$

12:      $C_{\text{tok}} \leftarrow [\,], G \leftarrow [\,]$            ▷ per-token confidence trace, and per-group averaged confidences

13:      $g\_sum \leftarrow 0, g\_len \leftarrow 0$            ▷ running stats for current group

14:      **for** $t = 1$ **to** $L$ **do**

15:          $\ell \leftarrow \pi_\theta.\text{NEXTLOGITS}(q, a, y); \quad \ell_{\text{ref}} \leftarrow \pi_{\text{ref}}.\text{NEXTLOGITS}(q, a, y)$

16:          $C_t \leftarrow \text{CONF}(\ell, \ell_{\text{ref}}, a)$            ▷ step confidence

17:          $C_{\text{tok}}.\text{APPEND}(C_t); \quad g\_sum \leftarrow g\_sum + C_t; \quad g\_len \leftarrow g\_len + 1$

18:          $LGC_t \leftarrow \min \big( (G = \varnothing\,?\,1 : \min(G)),\ g\_sum/g\_len \big)$      ▷ lowest group conf so far

19:          **if** $(LGC_t < \tau_a) \wedge (low \geq K \vee \text{MELTDOWN}(y))$ **then**

20:             $y.\text{APPEND}(\text{<EOS>}); $ **break**

21:          **else if** $(C_t < \tau_p) \wedge (p < M)$ **then**

22:             $y.\text{APPEND}(\text{<PAUSE>}); p \leftarrow p + 1$

23:             $h \leftarrow h \,\|\, \pi_\theta.\text{LATENTREASON}(q, a, y)$            ▷ generate until $\text{</PAUSE>}$

24:             **continue**

25:          **else**

26:             $tkn \leftarrow \text{SAMPLE}(\ell); \quad y.\text{APPEND}(tkn)$

27:             $low \leftarrow (C_t < \tau_p)\,?\,(low+1) : 0$

28:             **if** $tkn \in \{\text{<PAUSE>}, \text{</PAUSE>}\}$ **then**

29:                 $G.\text{APPEND}(g\_sum/g\_len); \quad g\_sum \leftarrow 0; g\_len \leftarrow 0$      ▷ close current group

30:             **end if**

31:             **if** $tkn = \text{<EOS>}$ **then break**

32:             **end if**

33:          **end if**

34:      **end for**

35:      $k \leftarrow \max(1, \lfloor \rho|C| \rfloor); \quad C_g \leftarrow \frac{1}{k} \sum_{c \in \min_k()} c$            ▷ get global confidence $C_g$

36:      **if** $(C_g < \tau_r) \vee \neg\mathcal{R}_{\text{cons}}(q, y)$ **then**            ▷ need explicit reflection

37:          $y_{\text{pre}} \leftarrow y; \quad (y_{\text{refl}}, y) \leftarrow \pi_\theta.\text{REFLECT}(q, a, y_{\text{pre}})$

38:      **else**

39:          $y_{\text{refl}} \leftarrow \varnothing, y_{\text{pre}} \leftarrow \varnothing$

40:      **end if**

41:      **return** $y, \{ y_{\text{pre}}, y_{\text{refl}}, \text{pauses} = p, \text{aborted} = (y[-1] = \text{<EOS>}) \}$

42: **end function**

---

